# Wind prevents cliff-breeding birds from accessing nests through loss of flight control

**Emily Shepard[1,2]\*, Emma-Louise Cole[1], Andrew Neate[3], Emmanouil Lempidakis[1], Andrew Ross[4]**

[1]Department of Biosciences, Swansea University, Swansea, United Kingdom; [2]Max Planck Institute for Animal Behaviour, Radolfzell, Germany; [3]Department of Mathematics, Swansea University, Swansea, United Kingdom; [4]School of Earth and Environment, University of Leeds, Leeds, United Kingdom

**Abstract** For fast-flying birds, the ability to respond to wind during landing is critical, as errors can lead to injury or even death. Nonetheless, landing ability, and its ecological significance, remain unstudied. We show that for auks, 60% of attempts to land at their cliff nests fail in a strong breeze (80% in near-gale winds). This is most likely because wind interferes with the ability to maintain flight control in the last phase of landing. Their extreme flight costs mean that the energetic penalty for multiple landing attempts is high. We propose that exposure, and ability to respond to, such conditions will influence the suitability of breeding habitat. In support of this (i) auk colonies appear to be orientated away from prevailing winds and (ii) landing success within colonies is higher on crowded ledges with more airspace for manoeuvring. More generally, the interplay between wind and flight capacities could impact breeding distributions across species and scales.
DOI: https://doi.org/10.7554/eLife.43842.001

\*For correspondence:
e.l.c.shepard@swansea.ac.uk

**Competing interests:** The authors declare that no competing interests exist.

## Introduction

The ability to fly profoundly affects the ecology of volant species: increasing the speed of travel (*Schmidt-Nielsen, 1972*), enabling animals to cross substantial barriers (*Hawkes et al., 2011*; *Schmaljohann et al., 2007*) and giving enhanced access to space across scales (*Kranstauber et al., 2015*). Indeed, flight underpins the most extensive and rapid annual migrations on the planet (*Kranstauber et al., 2015*; *Egevang et al., 2010*). In fact, the freedom that flight affords is such that the question of whether or not animals are able to access a given location within their range, or whether this ability is affected by environmental conditions, is almost never considered. Nonetheless, airflow characteristics can promote, impede or even prevent flight (*Shaffer et al., 2006*; *Ortega-Jimenez et al., 2014*; *Shepard et al., 2016b*; *Crall et al., 2017*), and this should have implications for an animal's ability to access key locations, when it comes to both moving through them and landing within them.

In aircraft, it is well recognised that landing is a period of high workload and relatively high risk. For instance, a Boeing review of worldwide commercial jet accidents between 2007 and 2016 showed that 48% of fatal accidents occurred during the final approach and landing (*Boeing, 2017*). Even for vehicles that move as fast as commercial jets, disturbed wind fields near runways can have a critical impact on flight control and safety, as highlighted by a European Aviation Safety Agency report (*van Es, 2012*). Yet in the biological literature, the process of landing has been examined mainly in relation to the sensory processes that guide it (*Baird et al., 2013*) and the biomechanics of force reduction on impact (*Bonser and Rayner, 1996*). How either of these may be affected by fluctuations in the wind field that are ubiquitous in the real-world flight environment, and the ecological

consequences of this, remain unknown. Indeed, in the only study that seems to have addressed this to date, Chang *et al* report that bumblebees (*Bombus impatiens*) landing on flowers shift from a multi-directional landing approach to a unidirectional approach upon the introduction of wind and, furthermore, that bees are unable to perform low-impact landings in windy conditions (*Chang et al., 2016*).

The forces of impact, and associated landing risks, will increase with animal mass and flight speed. Landing should therefore become more of a constraint for large animals, which also have lower available power (a factor that may be relevant if it is necessary to manoeuvre above the landing spot). Some of the largest birds, namely swans, overcome this by landing on water, where the impact can be reduced by extending the collision time and where momentum is transferred via the deformation of the water surface. For birds landing on solid substrates, regulation of their ground-speed is crucial during landing, as this will determine their impact with the landing surface. Airspeed, which is linked to lift production, must also be sufficient to remain airborne and maintain flight control. Both components must be modulated with respect to the wind speed. This is likely to be challenging in particular environmental conditions. For instance, it is reported that albatrosses have difficulty landing in low winds and that this can result in crash landings, broken bones and even death (*Cone, 1964*).

We quantify how landing ability varies in relation to wind speed at arguably the single most important location of all: the nest. Here, birds must be able to make repeated and safe landings, whatever the weather. We take colonially nesting seabirds as our study system. Cliff-nesting auks, including our study species, the common guillemot (*Uria aalge*) and razorbill (*Alca torda*), have among the highest wing loading recorded in birds (*Elliott et al., 2013*). This adaption for reduced diving costs (*Elliott et al., 2013*) means they have characteristically high flight speeds and low manoeuvrability. We therefore predict that the success of landing at the breeding cliffs will decline with the wind speed and/or turbulence experienced. We quantify the latter by combining direct observations of landing success and wind speed with outputs from computational fluid dynamics models, which are powerful tools with which to both visualise and estimate airflow characteristics around inaccessible places such as cliffs. We then develop a probabilistic model to assess how the ability to access nest sites, and the energetic costs of doing so, vary in relation to general wind conditions (i.e. a given mean wind speed and variance). Overall, this should provide new insight into the ways in which wind affects birds and other flying organisms, which is becoming increasingly important in light of the changing global wind conditions (*Weimerskirch et al., 2012*; *Young et al., 2011*).

## Materials and methods

### Study site and landing observations

Data were collected on Skomer Island, Pembrokeshire, UK (51.73611°N 5.29628°W). Here guillemots and razorbills form large, sympatric breeding colonies. Colony location and density were defined as follows: a digital elevation model (DEM, 0.5 m resolution, Lle Geo-Portal for Wales) was used to identify coastal cliffs, taken as regions with slopes > 20°. Cliffs were divided into breeding and non-breeding areas by digitising the 2015 Skomer Island breeding bird survey (*Stubbings et al., 2015*). Colonies were then defined as areas where birds were breeding, separated by distinct, unoccupied regions. The density of each colony was estimated by allocating the bird count to the associated cliff area, where the minimum height of the cliff was taken as 10 m ASL (to account for wave and tidal height [*Harris et al., 1997*]) extending to 15 m from the top of each cliff (using measurements made for the three largest colonies, E Shepard *unpubl. data*). We identified the densest colonies using a breakpoint in the density distribution. The mean orientation of each of these colonies was calculated, and a Rayleigh test was used to assess whether they were uniformly distributed. Analyses were conducted in ArcMap 10.5.1.

The landing attempts of guillemots and razorbills were observed over 26 days (28.4.2016–4.5.2017) at five breeding colonies, selected for being readily accessible and situated at different locations around the Island (SI, *Table 1*). Birds were assumed to be making landing attempts when they approached a cliff, usually ascending to it from below, steadily reducing the distance to the cliff until their ventral surface was orthogonally aligned with the cliff face. A landing was scored as

**Table 1.** The output of the best performing model.

High frequency wind speed measurements were obtained for 6140 observations and all models of landing success were run using this dataset. Height and ledge were included as factors.

| Parameter | Df | Estimate ± SE | p-value | F value | Deviance explained |
|---|---|---|---|---|---|
| wind | 1 | −0.62 ± 0.22 | <0.001 | 223.22 | 25.04 |
| ledge | 2 | | <0.001 | 72.51 | 16.27 |
| medium ledge | | −1.21 ± 0.12 | | | |
| small ledge | | −2.48 ± 0.15 | | | |
| turbulence | 1 | −0.81 ± 0.43 | 0.467 | 0.53 | 0.05 |
| species | 1 | | <0.001 | 122.57 | 13.75 |
| razorbill | | 1.10 ± 0.11 | | | |
| height | 3 | | 0.008 | 3.93 | 1.32 |
| lowest height | | −0.58 ± 0.19 | | | |
| wind * ledge | 2 | | <0.001 | 14.21 | 3.19 |
| wind*medium ledge | | 0.03 ± 0.04 | | | |
| wind*small ledge | | 0.22 ± 0.05 | | | |
| turbulence * ledge | 2 | | 0.015 | 4.23 | 0.95 |
| turb*medium ledge | | 0.95 ± 0.38 | | | |
| turb*small ledge | | 0.07 ± 0.50 | | | |

DOI: https://doi.org/10.7554/eLife.43842.002

successful if the bird touched down and stopped flying. An aborted attempt would begin in the same way, but birds would falter at the last moment and slip or fly away from the cliff. It was only possible to track the path of an individual over multiple landing attempts at one colony (High Cliff), as the return flight paths were partly obscured at all other observation points. The main dataset of landing attempts therefore refers to the success of focal individuals that were picked at random from all birds flying towards a breeding cliff. As such, our approach assumes that the overall dataset is not biased by a few individuals undertaking a large number of repeated attempts. We consider this reasonable given that landing attempts are short-lived events and that colonies are composed of hundreds of breeding pairs (SI *Table 1*). A smaller sample of birds was followed at High Cliff to count the number of repeat attempts under a given wind condition.

Landing ledges were grouped into the following categories according to the space available for landing (aligned with the platforms, ledges and niches identified in *Harris et al., 1997*): (1) relatively large, flat areas that were wider than an individual bird length in both horizontal dimensions, (2) ledges that were wider, but not deeper, than the length of an individual bird, and (3) areas that were less than the length of the bird in both horizontal dimensions. Each breeding cliff was also visually divided into four height bands of roughly 10 m, using landmarks to allow easy categorisation of the landing height.

Wind speed was recorded using a Kestrel anemometer positioned near the breeding cliff and the observer (SI, *Table 1*) at a height of 1.5 m AGL. This near-ground height was selected in order to assess conditions that birds might experience during the final phase of landing. The anemometer was positioned in the same location for all data collection sessions at a given colony and set to record once per minute.

## Modelling airflows

A modelling approach was used to (i) assess the extent to which wind speeds measured near observers varied from those on the breeding cliffs (it is logistically exceedingly difficult to measure wind speeds at the cliffs directly), (ii) relate near-ground values to the upwind/at sea condition, and (iii) estimate turbulence levels in relation to breeding colony and wind direction. Islands are particularly well suited to such modelling given that the upstream flow conditions are essentially uninterrupted. Airflows were modelled over Skomer using the open source computational fluid dynamics (CFD)

software OpenFOAM (openfoam.org). OpenFOAM has previously been validated and used for other atmospheric boundary flow problems, including the well-known Bolund test case (*Bechmann et al., 2011*) which involves modelling air flow over a small, steep island (*Cavar et al., 2016*). Our model domain was 5100×4950×1000 m, with a basic horizontal grid spacing of 25 m and a vertical spacing of 10 m. The height data for the lower boundary were taken from a DEM of the island at 5 m resolution (OS Terrain five dataset sourced from Digimap). The model mesh was fitted over the terrain and refined (2:1 refinement) up to twice near the lower surface using the OpenFOAM SnappyHexMesh tool. This gave the finest resolution near the surface as approximately 6.25 m in the horizontal and 2.5 m in the vertical plane. At the upwind boundary, a logarithmic wind profile was imposed, while outflow boundary conditions were applied at the downwind boundaries. The roughness length was set to 0.1 m. The model uses a k-ε turbulence closure scheme to find a steady state solution, which therefore provides both mean wind speed, U (m s$^{-1}$), and the turbulent kinetic energy, k (J m$^{-3}$). This is the kinetic energy associated with the turbulence rather than the mean wind, and is defined as:

$$k = \frac{1}{2}\rho\left(u'^2 + v'^2 + w'^2\right)$$

(1)

where u', v' and w' are the fluctuations about the mean in the three components of wind velocity (in the x, y and z directions) and ρ is the density (kg m$^{-3}$). Assuming u', v' and w' are all similar in magnitude (isotropic turbulence) then a 'typical' velocity perturbation, u, is given by u'$^2$ + v'$^2$ + w'$^2$ = 3 u$^2$. Substituting the equation for k and rearranging it gives:

$$u = \sqrt{((2/3)k/\rho)}$$

(2)

with u in m s$^{-1}$. In the model ρ is taken as 1 kg m$^{-3}$ and so can be neglected.

The strength of the turbulence was measured through the non-dimensional turbulence intensity, *I*:

$$I = u/U$$

(3)

Simulations were run for the following wind directions; N, NE, E, SE, S, SW, W and NW. The upwind wind profile was defined by the reference wind speed of 10 m s$^{-1}$ at a reference height of 20 m above the surface. Values of wind magnitude were normalised by the wind speed at an upwind reference point. For the W wind direction, additional simulations with input speeds of 5, 10 and 20 m s$^{-1}$ confirmed that both the normalised wind values and the turbulence intensities were independent of the input reference wind speed. Data were extracted 2 m normal to ground at the horizontal and vertical centre point of each of the five focal breeding cliffs, as well as at the associated observer positions.

## Statistical analysis

We used chi-squared tests to establish whether the two species differed in landing height and ledge size. Binomial generalised linear mixed effects models were applied to the landing success data using the R package LME4 and fitted with the bobyqa optimizer (*Bates et al., 2014*). Given that hypotheses could be developed for interactions between wind speed (as measured near the cliffs), turbulence levels, ledge type and species, an initial model was run with a 4-way interaction between these predictors, in order to identify which interactions featured regularly in the best fitting models. These interactions were included in the global model and model simplification was then performed using AIC values. The global model included two-way interactions between wind speed and ledge, and turbulence and ledge. Wind speed and turbulence values were centred to remove collinearity between the individual and interaction terms (*Schielzeth, 2010*). A variable combining day and colony was created and fitted as a random intercept in order to account for temporal and spatial autocorrelation in landing success that could occur due to wind direction (which would be altered from the mean condition in a particular way by the topography surrounding each colony). Tests for collinearity (using the CAR package [*Fox and Weisberg, 2011*]) and over-dispersion were run on the global model. Residual interpretation and goodness of fit tests were performed using the DHARMa package (*Hartig, 2017*) simulating residuals from 500 runs of the fitted model. Marginal and

conditional R$^2$ values were estimated using the MuMIn package (**Barton and Barton, 2018**). All analyses were conducted in R Studio Version 1.1.456 (**R Development Core Team, 2016**).

## Probabilistic modelling of landing failure

The statistical modelling related landing success to the wind speed recorded in that minute, as measured close to ground level. However, when wind speeds are considered over time, for example through the breeding season, or between years, it is the mean wind speed that is considered, and these records are made from anemometers stationed on weather buoys or at a greater altitude above ground level. We therefore developed a probabilistic model to (i) predict how landing success varies with mean wind speed, as measured further from the cliff (using the coefficients from the statistical model, as well as the airflow modelling), and (ii) derive the probability of landing in $n$ attempts, which was later used to estimate the metabolic cost of landing.

In both cases, a bird coming into land at a cliff effectively samples a distribution of wind speeds around a specified mean. We assumed that the instantaneous cliff wind speeds could be modelled as a Log Normal random variable. The probability distribution for the number of attempts taken to land successfully can then be derived from these parameters and the log odds of landing, with the latter taken from the best-fitting statistical model. This was converted to the probability of landing according to the mean wind speed over open water using constants derived from the airflow modelling.

Specifically, in our model, for a given situation (ledge, height, species) and fixed value of turbulence intensity, the probability of landing p=p(W) is a function of wind speed $W$ of the form

$$p(W) = \frac{a}{a + e^{bW}} \tag{4}$$

for constants $a$, $b$ determined by the statistical model. If a bird tries to land repeatedly until successful, the instantaneous wind speed at the cliff can be modelled as an independent Log normal random variable $W \sim LogNormal(m, s^2)$ where the parameters $m$ and $s$ are chosen so that the random variable $W$ has mean $U$ (mean wind speed at the cliff) and standard deviation $u$ (the root mean square of the turbulent fluctuations). That is, we choose $m$ and $s$ so that,

$$U = e^{m + \frac{s^2}{2}} \qquad u^2 = \left(e^{s^2} - 1\right) e^{2m + s^2} \tag{5}$$

Let $S$ denote the number of attempts required for a successful landing. Then $S$ has a geometric distribution with parameter $P$ given by the mean of $p(W)$ where W is the instantaneous wind speed at the cliff at the moment of each attempt to land. That is,

$$\mathbb{P}(S = n) = (1 - P)^{n-1} P \tag{6}$$

where,

$$P = \mathbb{E}(p(W)) = \int_0^\infty \frac{a}{(a + e^{bW})\sqrt{2\pi s^2 W^2}} e^{-\frac{(\ln W - m)^2}{2s^2}} \, \mathrm{d}W \tag{7}$$

## Results

### Wind data

Wind speeds at observer locations ranged between 0 and 11.6 m s$^{-1}$ during landing attempts. Similar wind speeds were recorded across study colonies, with median speeds ranging from 2.1 to 3.5 m s$^{-1}$ and reasonable maxima between 5.3 and 7.7 m s$^{-1}$.

Airflow models showed that there was a reduction in the wind speed close to the cliffs compared to the at-sea condition, with substantial areas of reduced wind speed in the lee of Skomer Island (**Figure 1A**) (**Figure 1—figure supplement 1**). In these downstream areas, the flow field tended to be more variable. Areas of high turbulence intensity also tended to occur in areas of reduced wind (**Figure 1C**).

Hourly records of at-sea wind speeds from the M5 wave buoy (51.41°N, −6.42 °W, where values are adjusted from 3.5 to 2 m ASL) showed that the median wind speed during the breeding season

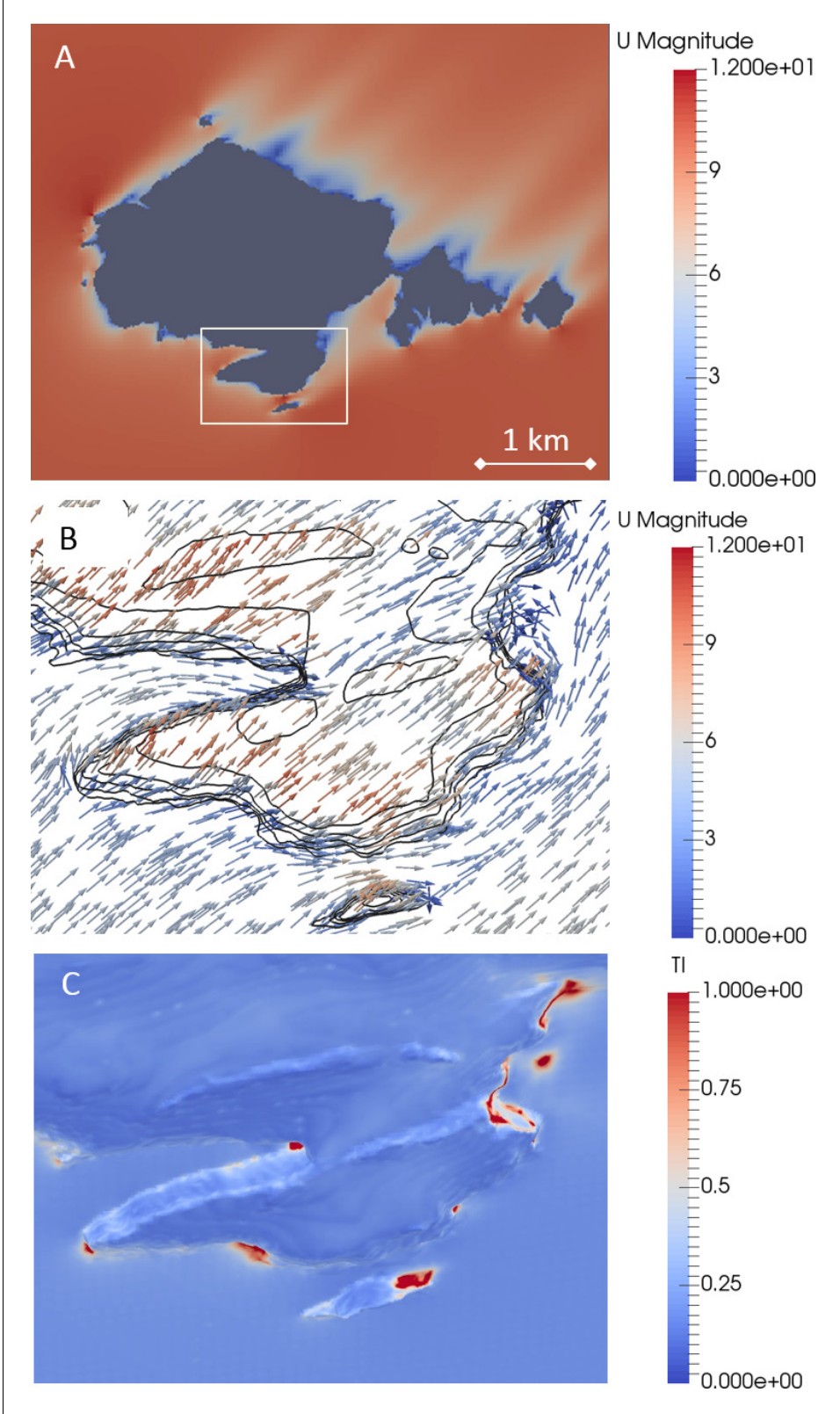

**Figure 1.** Airflows around Skomer Island (51.73611˚N 5.29628˚W), modelled with a SW wind. (**A**) A horizontal cross section of wind speed (m s$^{-1}$) at 10 m above sea level (which intersects the island, given in grey) shows the reduction in wind strength near the cliffs (see supplement 1). (**B**) The horizontal wind vectors within the inset in A, modelled at 2 m normal to the surface, and coloured according to the total wind speed (m s$^{-1}$). Wind is funnelled

*Figure 1 continued on next page*

*Figure 1 continued*
into the canyon on the left of the image (the Wick colony is located along the South side of this canyon), forcing birds to enter this area with a tailwind. (**C**) The turbulence intensity, TI, (a dimensionless ratio of the RMS of the turbulent wind fluctuations to the mean wind) at a distance of 2 m normal to the surface, within the inset shown in A. Typical values are ~0.1, so values of ~1 (red areas), indicate highly variable winds. Note these high values occur in areas with low mean winds (blue colours in *B*), so actual gust strength is low.
DOI: https://doi.org/10.7554/eLife.43842.003
The following figure supplement is available for figure 1:

**Figure supplement 1.** The mean wind speed at each colony is shown relative to the value at over the sea, where values < 1 indicate a reduction in wind speed close to the cliffs.
DOI: https://doi.org/10.7554/eLife.43842.004

(taken as March to August) was 6.1 m s$^{-1}$ (±2.1 IQR) and the reasonable maximum was 13.5 m s$^{-1}$ (actual maximum = 25.3 m s$^{-1}$).

## Landing data

Overall, 8623 landing attempts were recorded (guillemots n = 6140, razorbills n = 2483). Within this sample, birds were most likely to land on long thin ledges (42–52% of observations across study colonies) and least likely to land on the smallest ledges (11–33% of observations across study colonies). However, species differed in the ledges they selected, with razorbills landing on the smallest ledges more often than guillemots (Pearson's chi-squared test, $\chi2$ = 2639, df = 2, p<0.01, n = 8623) (*Figure 3—figure supplement 1*) and also landing on higher ledges ($\chi2$ = 813, df = 3, p<0.01, n = 8623).

High frequency wind speed measurements were obtained for 6140 observations. Statistical models of landing success were therefore run using this smaller dataset (guillemots n = 4257, razorbills n = 1883). The model with the lowest AIC score was the global model, which included interactions between wind speed and ledge, turbulence and ledge, as well as the main effects of species and height (AIC = 4687, conditional $R^2$ = 0.41, marginal $R^2$ = 0.36). The difference between the marginal and conditional $R^2$ demonstrates that the effect of site, and how this interacted with the daily wind condition, did not explain a disproportionate amount of variance. The next best approximating model (AIC = 4690, delta = 3) also included the interaction between wind speed and ledge, but not the interaction between turbulence and ledge, which was dropped from this model (conditional $R^2$ = 0.47, marginal $R^2$ = 0.31). There was no evidence for overdispersion in the global model (ratioObsExp = 0.941, p=0.999) or collinearity between predictors, and residual plots in DHARMa provided no evidence of heteroscedasticity.

Wind speed was the variable that explained the greatest amount of variation in landing success, with the probability of an auk landing on its breeding cliff decreasing with wind speed (estimate = −0.62 ± 0.05, p<0.01, df = 6139, 1, F = 223.22, expl. dev. 25.04%). Landing success was close to 100% in wind-still conditions, decreasing slowly with winds up to ~4 m s$^{-1}$, before decreasing more rapidly to a predicted success rate of <20% in winds of 8 m s$^{-1}$ (*Figure 2*). The probability of landing decreased with ledge area, being lowest for the smallest ledges (expl. dev. 16.27%, see *Table 1* for all parameter estimates and details of model outputs). Species also explained a substantial proportion of variation in landing ability (expl. dev. 13.75%), with razorbills being more likely to land successfully. There were significant interactions between ledge size and wind speed (expl. dev. 3.19%) and ledge and turbulence (expl. dev. 0.95%). Interestingly, the probability of landing did not appear to vary with turbulence levels (as a main effect), which may be in part because turbulence levels were highest in areas with relatively low horizontal wind speeds (*Figure 2*). Finally, birds were more likely to land successfully on higher ledges, although the explanatory power of landing height was extremely low (expl. dev. 1.32%).

## Probabilistic models of landing failure

The relationship between landing success and at-sea wind speeds was qualitatively similar to the statistical model, although the success was higher for any at-sea wind speed due to the reduction in wind speed that occurs close to the cliffs (*Figure 2*).

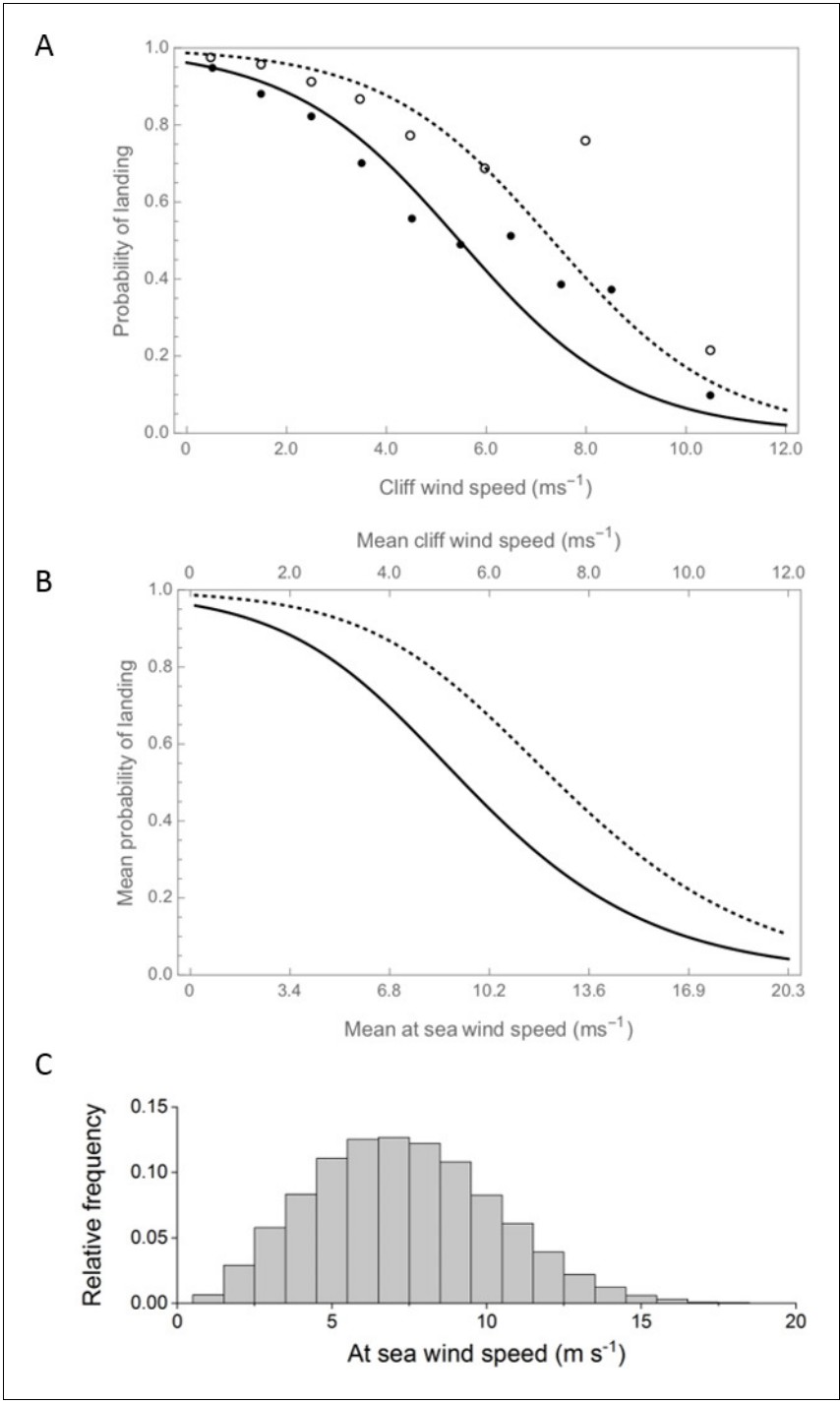

**Figure 2.** Landing success decreases with increasing wind speed for both guillemots (solid line) and razorbills (dashed line). (**A**) The probability of landing (derived from the statistical model), declines to ~0.1 in winds of 10 m s$^{-1}$. Binned raw data are shown for both species (guillemots as filled circles, razorbills as open circles; data are grouped with n $\geq$ 30 observations per bin). As success also varies with ledge type, both the model output and raw data refer to birds landing on long narrow ledges. (**B**) The probability of landing according to the mean, at-sea wind speed, as derived from the probabilistic model (also for long narrow ledges and a TI value of 0.2). The difference between the x-axes indicates the increase in wind speed over open water, compared to near the cliffs, as estimated using airflow model outputs averaged across all wind directions. (**C**) The distribution of at-sea wind speeds across the breeding season (for 2005–2018, recorded at the M5 wave buoy and reduced to 2 m ASL).
DOI: https://doi.org/10.7554/eLife.43842.005

*Figure 2 continued on next page*

*Figure 2 continued*

The following source data is available for figure 2:

**Source data 2.** The raw data on landing observations, along with associated data on species, ledge type and wind speed (see also 'parameter definitions').

DOI: https://doi.org/10.7554/eLife.43842.006

The number of attempts required to land increased with wind speed for both species (*Figure 3*). All birds were predicted to land within 3–8 attempts for low wind speeds (4 m s$^{-1}$), depending on species and ledge size. In winds of 10 m s$^{-1}$, which are close to the reasonable maximum speeds expected during the breeding season (*Figure 2C*), guillemots may need up to 20 attempts to land, even on the largest landing platforms, whereas razorbills are predicted to land in roughly half the number of attempts.

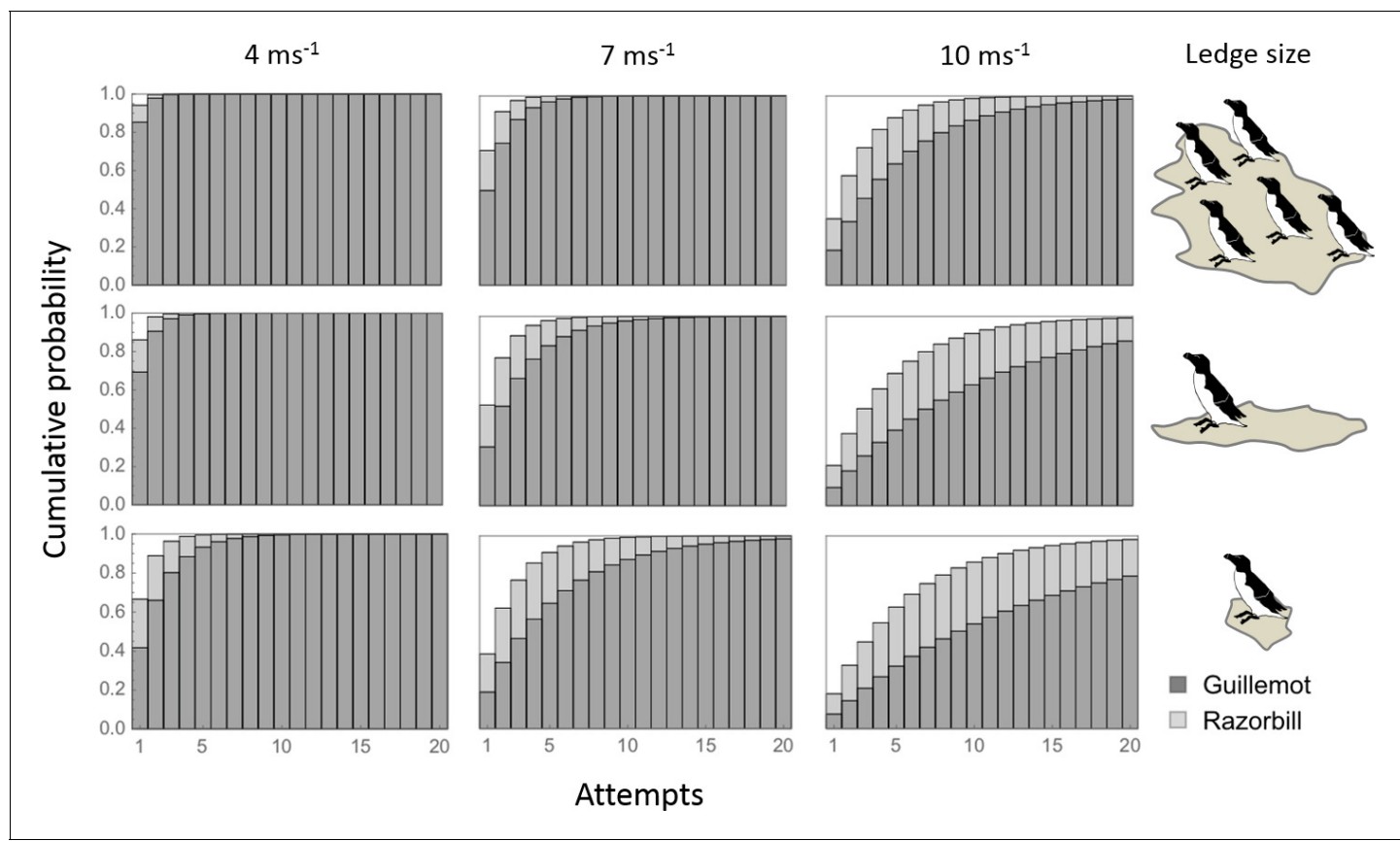

**Figure 3.** The cumulative probability of landing according to wind speed, ledge type and species. Seasonal wind speeds near the breeding cliffs will vary with wind direction and colony location (*Figure 1—figure supplement 1* ). For the prevailing SW wind direction, median wind speeds across the breeding season are predicted to be ≤7 m s$^{-1}$ (first two columns). Upper quartile speeds are predicted to be ≤ 9 m s$^{-1}$ (third column), and reasonable maxima ≤ 16 m s$^{-1}$.

DOI: https://doi.org/10.7554/eLife.43842.007

The following figure supplements are available for figure 3:

**Figure supplement 1.** Observations of landings grouped according to ledge size for each species.

DOI: https://doi.org/10.7554/eLife.43842.008

**Figure supplement 2.** The largest guillemot colony on Skomer; the Wick.

DOI: https://doi.org/10.7554/eLife.43842.009

## Colony orientation

The cliff area occupied by breeding auks was 76,673 m$^2$, compared to 122,302 m$^2$ of unoccupied cliff. While the 11 densest colonies appeared to be orientated in a range of directions bar those facing the prevailing SW wind, a Rayleigh test failed to reject the null hypothesis that the orientations were not uniformly distributed (Z = 0.314, p=0.346, df = 10) (*Figure 4*). In contrast, a number of unoccupied sites had a south-westerly orientation (*Figure 4B*).

## Discussion

Like many colonially nesting birds, auks coming in to land on their breeding cliffs must moderate their movements in relation to nearby neighbours, a partner already at the nest site and, critically, their egg or chick. The fact that auks manage to land (and breed) on ledges so narrow that their tails hang over the edge (mean density = 20 per m$^2$ [*Harris and Birkhead, 1985*]), is testament to their flight capacity. However, we show that even moderate winds can upset this delicate balancing act. In our study, the ability to land was significantly impacted by wind speeds on the approach to the breeding cliffs. When translated into the probability of landing in a mean wind condition, birds only landed reliably (p(landing)>0.9) in calm to light air, equivalent to at-sea winds of $\leq$2 m s$^{-1}$. The probability of landing successfully fell to 0.4 in a strong breeze and to 0.2 in near gale conditions (11 and 15 m s$^{-1}$ over open water respectively). Thus, even in conditions that are not sufficiently severe to be categorised as either gales or storms, wind can act as an invisible barrier, affecting the ability of cliff-nesting birds to access their chicks. The outcome is that the need to avoid terrestrial predators (as inferred from their cliff-nesting habit) has driven birds to select places they themselves are unable to access in some conditions. This occurs despite the fact that wind speeds close to the cliffs are lower than those over open water and that birds in our study appear to select breeding cliffs that are oriented away from the prevailing wind direction.

During migratory and commuting flights, diverse animals vary when and where they fly in relation to wind conditions, selecting airflows that are beneficial for flight costs or flight control (*Kranstauber et al., 2015*; *Shaffer et al., 2006*; *Shepard et al., 2016a*; *Sapir et al., 2014*). In stark contrast, decisions about where to breed cannot be changed after the initial investment at the start of a breeding season. Cliff-nesting auks therefore have no choice but to attempt to return to this fixed place, even in difficult conditions, in order to provision their young and relieve their partner. Animals including bees have been shown to vary their landing trajectory in relation to the wind

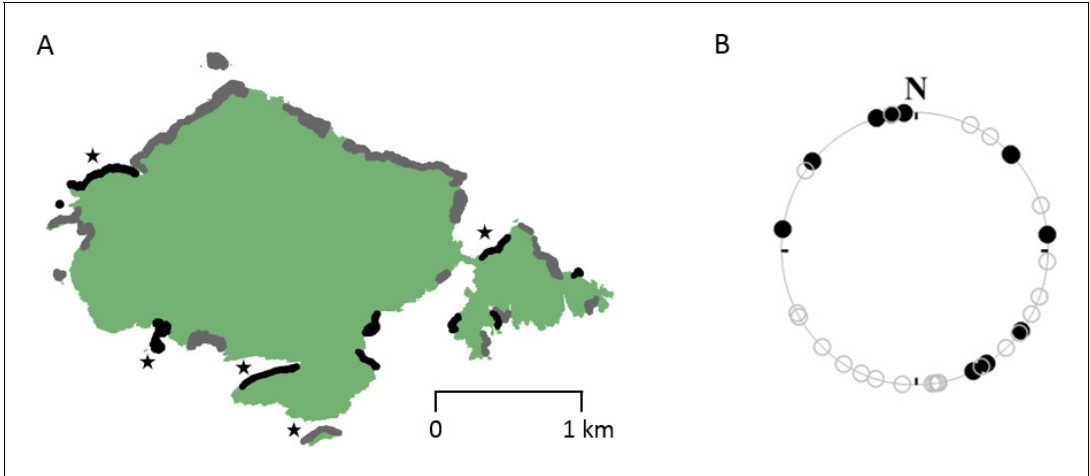

**Figure 4.** The distribution and orientation of guillemot breeding colonies on Skomer Island (digitised from the 2015 breeding bird survey [*Stubbings et al., 2015*]). (**A**) Breeding colonies (grey and black regions) appear to be distributed all around Skomer, however, bearings of the 11 densest colonies (marked in black), given in (**B**), show that while colonies appear uniformly distributed overall (Z = 0.314, p=0.346, df = 10, Rayleigh test), none are oriented towards the prevailing south-westerly wind direction, despite the availability of cliff habitat (unoccupied sections are indicated with open circles). Study colonies are indicated with stars.
DOI: https://doi.org/10.7554/eLife.43842.010

vector (*Chang et al., 2016*). However, we show that the extent that cliff-nesting birds can vary their approach trajectory with wind direction may be limited by the topography of the surrounding area (*Figure 2*). For instance, in prevailing SW winds, our airflow modelling shows that birds landing at the largest guillemot colony on our study site (the Wick, Skomer Island *Figure 2*), are forced to fly parallel to the cliff, with a substantial tailwind component, before making their final approach in a crosswind. Consequently, the ability of cliff-nesting auks to land in these conditions at all is remarkable.

In high winds, adults can choose to delay foraging, and indeed, there is evidence that a range of auk species do reduce feeding rates in windy conditions (*Konarzewski and Taylor, 1989*; *Birkhead, 1976*; *Elliott et al., 2014*). However, delaying foraging for long periods is a poor option, particularly when provisioning small chicks. The alternative is to perform multiple landing attempts. The energetic cost of an additional landing can be estimated using the median time taken for completion of one landing circuit, as measured at one of our study colonies (34 s, range 11–58 s, n = 30), and the costs of level flight as measured by *Elliott et al. (2013)*. This gives 4.9 kJ per landing loop. To put this into context, the gross calorific value of one lesser sandeel (*Ammodytes marinus*) of a size consumed by guillemots (*Hislop et al., 1991*; *Pearson, 1968*) would be equivalent to the cost of ~6 attempts. How often are guillemots likely to have to pay this price? Given that wind speed varies through time, the number of attempts required for a successful landing will vary with both the mean wind speed and the variance around this mean. Our probabilistic model shows that the probability of a guillemot landing in six attempts decreases from one in a gentle to moderate breeze, to 0.4 in near gale conditions (representing the median and routine maximal wind speeds during the breeding season respectively). As guillemots typically perform 3–4 provisioning trips per day (*Thaxter et al., 2009*), the cumulative daily cost of multiple landing attempts per trip could be substantial.

Despite this, adults are still likely to perform multiple attempts rather than risk landing without proper flight control, as the territory size is so small that any error during the parental changeover almost invariably results in the loss of the egg or chick (*Kokko et al., 2004*). In Brünnich's guillemot colonies (*Uria lomvia*), adult changeovers have also been identified as periods when eggs and young are vulnerable to gull predation (*Gilchrist et al., 1998*, cf. *Ashbrook et al., 2008*). The negative effects of high winds could be two-fold for these auks; impeding their own flight capacities, whilst simultaneously promoting the capacity of gulls to fly and manoeuver close to the narrow breeding ledges (*Gilchrist and Gaston, 1997*).

In our study, the size of the landing platform was the most important parameter to influence landing success after wind strength. Paradoxically, this is unlikely to be due to the physical area of the platform itself, as large platforms are typically crowded (which can produce further complications as neighbouring birds can reach up to peck incoming individuals, E Shepard, *pers. obs*). Consequently, the area of rock available for landing may not vary substantially across platform sizes. Larger platforms do, however, have greater available airspace above them, which could be important in allowing birds to manoeuvre in the final phase of landing and therefore achieve greater flight control in stronger winds.

The ability to land more readily in some locations than others represents an aspect of site quality that has not previously been considered (*Harris et al., 1997*; *Kokko et al., 2004*; *Birkhead et al., 1985*), beyond a study in 1964 (*Cone, 1964*), suggesting that albatrosses choose to nest on the windward side of islands in order to facilitate landing and take-off. Our data show that within a given colony, guillemots nesting on large ledges will experience benefits beyond the extra protection from aerial predators that is afforded by having a greater number of neighbours (*Birkhead, 1977*). Quality will also vary between different cliff sites, with those providing greatest protection from the wind presumably offering more reliable conditions to land in. In support of this, our preliminary data suggest that colonies are orientated in most directions bar those that face into the prevailing wind direction. However, more substantial analyses are needed to ascertain which components of the wind field are most important when it comes to site selection. Even then, it may be difficult to disentangle the ultimate drivers of site selection, as shelter from strong winds could also be important in reducing exposure to low temperatures, or the possibility of young being swept off ledges by strong winds or associated wave action (*Bonter et al., 2014*; *Høyvik Hilde et al., 2016*). Furthermore, shelter is likely to improve the ability of fledglings to jump and reach the sea at the end of the season (*Gilchrist and Gaston, 1997*). Notwithstanding this, the relevance for the present study is that we

see surprisingly high rates of landing failure despite the fact that birds appear not to be breeding in the most exposed sites.

Habitat suitability and quality are therefore likely to vary according to the prevailing conditions, how these interact with local topography (which will determine exposure levels and the ability of individuals to adjust the direction of their final approach), and fine-scale ledge characteristics that affect the extent that birds can manoeuvre above the landing spot. Nonetheless, the importance of these factors will also vary with species, with a given area being simultaneously accessible to some species and inaccessible to others. In our study, guillemots were much more susceptible to the effects of wind than razorbills. In fact, in the strongest winds we recorded, the probability of guille-mots landing on small ledges in 10 attempts was just 0.55, compared to 0.85 in razorbills. The mor-phology of guillemots and razorbills is extremely similar, but guillemots are larger and out-compete razorbills for access to the bigger platforms (*Linnebjerg et al., 2013*). Indeed, in this study, razorbills landed on the smallest ledges more often than expected, with the reverse being true for guillemots. If ledge type were the sole determinant of landing ability, we would therefore expect razorbills to have inferior landing success. The increased manoeuvrability of razorbills (due to their lower wing loading and hence flight speed) is likely to be the reason why the opposite is true, as razorbills should be able to turn more tightly, and hence respond appropriately to wind in the last phase of landing. The chances of wind having a substantial impact on the energetic costs of landing are there-fore far greater in guillemots, which could, in turn, influence their motivation to compete for the larger ledges.

## Conclusions

Decisions such as when and where to fly are fundamentally linked to wind conditions, because flight speed and flight costs vary with the wind vector (*Hedenstrom and Alerstam, 1995*). This affects population processes in a number of ways, influencing the most cost-effective migration routes (*Kranstauber et al., 2015*), species distributions (*Davies et al., 2010*) and shift durations in breeding birds (*Weimerskirch et al., 2012*). Our study provides evidence of an additional mechanism by which wind can affect birds, effectively preventing cliff-nesting auks from landing in high winds. The issue of how to land safely in windy conditions is pertinent for a wide range of species, as the diver-sity of several seabird groups increases with latitude and wind speed (*Davies et al., 2010*; *Cairns et al., 2008*). Low manoeuvrability may represent a constraint in this regard for auks – and one that is likely to be most critical in Brünnich's guillemots, which have a wing loading some 20% higher than common guillemots (*Elliott et al., 2013*). However, other aspects of flight control, such as stability, could be limiting for species with lower wing loading. This opens up a great many ques-tions on how birds of different morphologies and flight capacities respond to strong and potentially variable airflows during landing, and the extent to which this modulates patterns of space use in the breeding season. Indeed, we expect that inter-specific variation in flight capacities may help explain why there is such clear distinction between the areas that different seabird taxa occupy on offshore islands. In general, we predict that cliff-nesting species with low manoeuvrability should select shel-tered sites for breeding. The preference for low wind speeds may also affect the timing of breeding, with birds postponing their return to the breeding cliffs, or their lay date, if these periods coincide with persistent strong winds (*cf*. *Wanless et al., 2009*). This highlights the importance of monitoring the timing of strong winds during the breeding season, as well as the frequency. Overall, this is perti-nent as wind regimes are changing (*Young et al., 2011*) and there is a need to establish a compre-hensive framework to understand, and ultimately predict, which species are likely to be affected, and how (*Lewis et al., 2015*).

## Acknowledgements

We are grateful to the Wildlife Trust of South and West Wales and Skomer wardens Bee Büche and Eddie Stubbings for supporting data collection. This project was supported by the European Research Council under the European Union's Horizon 2020 research and innovation program Grant 715874 (to ELCS). The OpenFoam modelling was undertaken on ARC3, part of the High Perfor-mance Computing facilities at the University of Leeds, UK. Wind data from the Coastguard lookout station at Wooltack point were kindly provided by Natural Resources Wales. Harry Read and Jess Ware assisted with data collection. Luca Börger and Rory Wilson provided valuable statistical advice

and comments on an early draft respectively. Finally, we thank Kyle Elliot, Yuuki Watanabe and Gil Bohrer for their comments, which led to substantial improvements in the manuscript.

## Additional information

### Funding

| Funder | Grant reference number | Author |
| --- | --- | --- |
| H2020 European Research Council | 715874 | Emily Shepard |

The funders had no role in study design, data collection and interpretation, or the decision to submit the work for publication.

### Author contributions

Emily Shepard, Conceptualization, Formal analysis, Funding acquisition, Methodology, Writing— original draft, Project administration, Writing—review and editing; Emma-Louise Cole, Investigation, Methodology; Andrew Neate, Andrew Ross, Conceptualization, Formal analysis, Methodology, Writing—original draft, Writing—review and editing; Emmanouil Lempidakis, Formal analysis, Methodology, Writing—review and editing

### Author ORCIDs

Emily Shepard https://orcid.org/0000-0001-7325-6398
Andrew Ross https://orcid.org/0000-0002-8631-3512

### Decision letter and Author response

Decision letter https://doi.org/10.7554/eLife.43842.016
Author response https://doi.org/10.7554/eLife.43842.017

## Additional files

### Supplementary files

• Source code 1. Models of landing success.
DOI: https://doi.org/10.7554/eLife.43842.011

• Supplementary file 1. Summary information for the breeding cliffs where landing data were collected. Numbers of guillemots (GM) are taken from the 2015 Skomer Island breeding bird survey (*Stubbings et al., 2015*).
DOI: https://doi.org/10.7554/eLife.43842.012

• Supplementary file 2. Definitions of parameters within "Skomer Landings" data.
DOI: https://doi.org/10.7554/eLife.43842.013

• Transparent reporting form
DOI: https://doi.org/10.7554/eLife.43842.014

### Data availability

Landing observations are included in the supporting files. The raw data for the GLMM (Source data 1) and the GLMM code (Source code 1) have been uploaded as additional data files (csv and R files respectively). The parameters listed in the raw data file are defined in an accompanying txt file (Supplementary file 2).

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
