## [Decision Letter]

Thank you for submitting your article "Wind prevents cliff-breeding birds from accessing nests through loss of flight control" for consideration by *eLife*. Your article has been reviewed by four peer reviewers, including Christian Rutz as the Reviewing Editor and Reviewer #1, and the evaluation has been overseen by Ian Baldwin as the Senior Editor. The following individuals involved in review of your submission have agreed to reveal their identity: Kyle Elliot (Reviewer #2); Yuuki Watanabe (Reviewer #3); Gil Bohrer (Reviewer #4).

The reviewers have discussed the reviews with one another and the Reviewing Editor has drafted this decision to help you prepare a revised submission.

Summary:

This study is one of the first to examine the dynamic component of flight in wild birds, and as such represents a notable advance. It brings an exciting topic into focus, through effective combination of field observations and airflow modelling, and will likely inspire future work on a wide range of taxa.

Essential revisions:

1) Presentation: While the reviewers enjoyed the lucid writing style, there was broad agreement that the presentation of the material could be improved. Please use separate Materials and methods, Results and Discussion sections (in that order) to structure the narrative better, and: (a) include sample sizes for all datasets and analyses (number of colonies, birds/landing attempts observed etc.); (b) provide more comprehensive summaries of the underlying data (including landing success/failure data for each species); (c) report results separately throughout for the two study species (guillemots and razorbills); and (d) clarify which analyses, and inferences, are based on observational data and which ones refer to your modelling work.

2) Sampling: Your work is based on two samples of colonies from a single island (Skomer): a set of 19 colonies for assessing orientation patterns, and a subsample of five colonies for observing birds' landing attempts. Please provide more information on how these samples were selected. You say that the 19 colonies were the "main" (subsection “Data collection”) or "largest" colonies (Figure 3 legend). How were these terms defined here (spatial extend or number of nests), what was your size criterion, and why did you not sample across the full range of colony sizes (from small to large), or include all local colonies? This is important, as you use this sample to conduct formal statistical analyses (Figure 3). In fact, given the good availability of suitable data for auks, the reviewers felt it would be important to replicate distribution analyses across different islands, to rule out the possibility that patterns are driven by factors other than wind conditions (such as access to preferred foraging sites). Finally, does your subsample of five colonies (subsection “Data collection”, first paragraph; Supplementary file 1) adequately capture the full variation of environmental conditions experienced by birds on Skomer? It would also be useful to know sample sizes – in terms of sites rather than individuals – for each wind speed measurement (e.g., did all the data on the windiest day come from a single site?). Finally, if different colonies are at different locations with different ledge sizes, statistical analyses would need to account for this.

3) Modelling: For your wind speed distribution, you use a normally distributed variable. Wind speed is only positive. Wind velocity at each of the 3 spatial components can be negative, but the bird does not care if the wind goes left or right – that makes it just as hard to land. You assume, correctly, that the birds are affected by the speed of the wind vector, and the speed [sqrt(sum(ui^2)) |i=1:3] is a positive property. Please use log normal distribution for W, which presumably will affect Equation 5.

---

## [Author Response]

Essential revisions:1) Presentation: While the reviewers enjoyed the lucid writing style, there was broad agreement that the presentation of the material could be improved. Please use separate Materials and methods, Results and Discussion sections (in that order) to structure the narrative better, and: (a) include sample sizes for all datasets and analyses (number of colonies, birds/landing attempts observed etc.); (b) provide more comprehensive summaries of the underlying data (including landing success/failure data for each species); (c) report results separately throughout for the two study species (guillemots and razorbills); and (d) clarify which analyses, and inferences, are based on observational data and which ones refer to your modelling work.

We have changed the format so there are separate Results and Discussion sections, and these are preceded by the Materials and methods. The new format allows for more detail in the Results section, including sample sizes (which now feature most in the first and second paragraphs of the new Results section) and summaries of the underlying data. Table 1 has also been expanded to provide more detail on the statistical model. Further changes relating to points (b), (c) and (d) are detailed below.

2) Sampling: Your work is based on two samples of colonies from a single island (Skomer): a set of 19 colonies for assessing orientation patterns, and a subsample of five colonies for observing birds' landing attempts. Please provide more information on how these samples were selected. You say that the 19 colonies were the "main" (subsection “Data collection”) or "largest" colonies (Figure 3 legend). How were these terms defined here (spatial extend or number of nests), what was your size criterion, and why did you not sample across the full range of colony sizes (from small to large), or include all local colonies? This is important, as you use this sample to conduct formal statistical analyses (Figure 3).

Apologies that this was not clear before. The five colonies where observations were made were selected as they are distributed fairly evenly around the island, which we felt was important as they might experience different wind conditions on a given day. Colonies also had to be readily accessible to obtain observations from multiple locations within a day for fieldworkers travelling on foot. The study colonies did include a range of sizes and we have reworded the Materials and methods where colony selection is discussed (subsection “Study site and landing observations”, second paragraph) and where it implied that we selected the 5 largest colonies. The number of birds in these colonies is given in Supplementary file 1.

We also revisited the section where colonies in general are defined and have reworked this to provide a much more rigorous definition of colonised areas (subsection “Study site and landing observations”, first paragraph). This has also resulted in a new version of Figure 4, which now includes the spatial extent of all colonies.

In fact, given the good availability of suitable data for auks, the reviewers felt it would be important to replicate distribution analyses across different islands, to rule out the possibility that patterns are driven by factors other than wind conditions (such as access to preferred foraging sites).

We feel it is unlikely that the landing performance of birds could be influenced by possible differences in foraging sites, given the importance of wind in the statistical model, and that wind seems to be most critical in the last phases of the landing. Our main aim here is to introduce the idea that exposure to wind could have implications for breeding distributions and our approach is preliminary, as it does not account for factors such as colony size. We appreciate that distribution data are available for other islands, however, a thorough investigation of the factors driving the distribution of colonies, even within Skomer, is beyond the scope of this study as it would require detailed testing of different hypotheses. We have therefore adjusted the wording in the Results and Discussion to reflect this (subsection “Probabilistic models of landing failure”, Discussion, first and sixth paragraphs).

Finally, does your subsample of five colonies (subsection “Data collection”, first paragraph; Supplementary file 1) adequately capture the full variation of environmental conditions experienced by birds on Skomer? It would also be useful to know sample sizes – in terms of sites rather than individuals – for each wind speed measurement (e.g., did all the data on the windiest day come from a single site?). Finally, if different colonies are at different locations with different ledge sizes, statistical analyses would need to account for this.

The precise wind conditions that birds experience as they land will be determined by fairly complex interactions between the wind direction, its variation in time, and its variation in space. The latter will be determined by how the wind interacts with the local topography. We tried to capture this variation in a statistical sense by creating the random variable Day|Site, to cover the interaction between wind conditions on a given day, and the topography.

In terms of experimental design, our study colonies reflect much of the range of bearings across which all colonies are orientated (apart from SE), which is now clarified in the Materials and methods (subsection “Study site and landing observations”). However, we appreciate that this is a simplistic measure of the conditions that will be experienced within a colony as a whole.

Finally, a good range of wind speeds was measured at all colonies and birds landed on the different ledge sizes in similar proportions across the colonies. The summary statistics for this are now reported in the first and fourth paragraphs of the Results.

3) Modelling: For your wind speed distribution, you use a normally distributed variable. Wind speed is only positive. Wind velocity at each of the 3 spatial components can be negative, but the bird does not care if the wind goes left or right – that makes it just as hard to land. You assume, correctly, that the birds are affected by the speed of the wind vector, and the speed [sqrt(sum(ui^2)) |i=1:3] is a positive property. Please use log normal distribution for W, which presumably will affect Equation 5.

We have redone the modelling using a log normal distribution. As a result, we have also changed the text in the Materials and methods section “Probabilistic modelling of landing failure” and redone Figures 2 and 4. The change in distribution had little impact on the nature of the relationships in the figures (as very few wind speeds were negative when the normally distributed variable was used), and consequently no substantial changes were required to the text elsewhere.